# Ca^2+^ Dynamics of Gap Junction Coupled and Uncoupled Deiters’ Cells in the Organ of Corti in Hearing BALB/c Mice

**DOI:** 10.3390/ijms241311095

**Published:** 2023-07-04

**Authors:** Louise Moysan, Fruzsina Fazekas, Adam Fekete, László Köles, Tibor Zelles, Eszter Berekméri

**Affiliations:** 1Department of Zoology, University of Veterinary Medicine Budapest, H-1078 Budapest, Hungary; louise.moysan35@gmail.com (L.M.);; 2Program in Neurosciences and Mental Health, The Hospital for Sick Children, Toronto, ON M5G 1X8, Canada; 3Department of Oral Biology, Semmelweis University, H-1089 Budapest, Hungary; 4Department of Pharmacology and Pharmacotherapy, Semmelweis University, H-1089 Budapest, Hungary; 5Laboratory of Molecular Pharmacology, Institute of Experimental Medicine, H-1083 Budapest, Hungary

**Keywords:** single-cell electroporation, Ca^2+^ imaging, DCs, calcium signalling, tonotopy, gap junction, CBX, octanol, computational model

## Abstract

ATP, as a paracrine signalling molecule, induces intracellular Ca^2+^ elevation via the activation of purinergic receptors on the surface of glia-like cochlear supporting cells. These cells, including the Deiters’ cells (DCs), are also coupled by gap junctions that allow the propagation of intercellular Ca^2+^ waves via diffusion of Ca^2+^ mobilising second messenger IP_3_ between neighbouring cells. We have compared the ATP-evoked Ca^2+^ transients and the effect of two different gap junction (GJ) blockers (octanol and carbenoxolone, CBX) on the Ca^2+^ transients in DCs located in the apical and middle turns of the hemicochlea preparation of *BALB*/*c* mice (P14–19). Octanol had no effect on Ca^2+^ signalling, while CBX inhibited the ATP response, more prominently in the middle turn. Based on astrocyte models and using our experimental results, we successfully simulated the Ca^2+^ dynamics in DCs in different cochlear regions. The mathematical model reliably described the Ca^2+^ transients in the DCs and suggested that the tonotopical differences could originate from differences in purinoceptor and Ca^2+^ pump expressions and in IP_3_–Ca^2+^ release mechanisms. The cochlear turn-dependent effect of CBX might be the result of the differing connexin isoform composition of GJs along the tonotopic axis. The contribution of IP_3_-mediated Ca^2+^ signalling inhibition by CBX cannot be excluded.

## 1. Introduction

The cochlea is found in the inner ear. It consists of a bony structure with a characteristic spiral shape which can be divided into apical, middle and basal regions in mice [1,2]. The structural and mechanical arrangements along the cochlear turns enable frequency discrimination and ensure that the auditory nerve fibres fire in a frequency-coded manner called tonotopy [2,3,4].

In the organ of Corti, hair cells are the primary auditory receptor cells [5]. They are surrounded by supporting cells, including DCs [6,7,8] which provide physical and metabolic support to the outer hair cells (OHCs) [6,9]. In addition, they help maintain ion homeostasis in the endolymph via active K^+^ uptake [6,10], and studies have demonstrated that they exhibit functions like glial cells [6,8]. The function and properties of the mechanosensory hair cells are widely investigated with the aim of studying their connection to impaired hearing function and potentially developing therapeutic treatment [11]. However, fewer studies focus on the physiological roles and cellular mechanisms of the inner ear’s supporting cells, including DCs. Supporting cells are drawing increasing attention, and it is believed their physiology could influence hearing mechanisms, like threshold sensitivity and hair cell survival [6,12,13].

In the mammalian cochlea, Ca^2+^ ions are implicated in all steps of the transduction process, mechanoelectrical transduction function, neurotransmitter release, cochlear amplification, ion homeostasis in the endolymph and developmental processes [8,14,15,16,17]. Two types of purinergic receptors are expressed in DCs: ionotropic P2X and metabotropic P2Y receptors. Both receptor types induce intracellular Ca^2+^ increase in the cytoplasm when stimulated by extracellular ATP [1,2,18]. These cells are also coupled via gap junction channels, which are primarily made up of connexins 26 (Cx26) and connexins 30 (Cx30) [10]. It allows the intercellular transfer of Ca^2+^-mobilizing second messengers IP_3,_ which may play a role in the K^+^ recycle function [6,10,17,19,20,21,22]. To investigate the gap junction coupling mechanism, octanol and CBX can be used to block non-selectively the connexin channels [19,23,24,25]. In addition to its gap junction blocking effect, CBX is also thought to block IP_3_-mediated Ca^2+^ release by inhibiting IP_3_R activity and voltage-gated Ca^2+^ channels [25,26]. 

Cellular-level investigation in the cochlea is mostly carried out on young mice (P1–2) before the temporal bone calcifies. However, mice are born deaf, and their hearing organ does not mature until the end of the second post-natal week [2,8,27,28]. In addition, its complex morphology makes it challenging to gain usable data from the sensory organ of hearing. Using a Ca^2+^ imaging experiment, we investigated the ATP-induced Ca^2+^ transients of DCs in hemicochlea preparation of hearing mice. In the presence of gap junction blockers (octanol and CBX), we stimulated the cells and measured changes in cellular reactivity to explore their coupling strength and the effect of gap junction blockers on the transients. We also developed a mathematical model that simulates the calcium patterns in DCs of the apical and middle regions. Our model is an extended model considering cytosolic Ca^2+^ concentration, the fraction of active inositol trisphosphate receptors (IP_3_R) and IP_3_ dynamics as variables. Since no model has been developed for DCs, we based our model on Taheri et al. [29,30] and De Pittà et al. [31] models that both simulate Ca^2+^ dynamics in astrocytes. We also explore the gap junction coupling effect employing Wu et al. [32] model. Since CBX may also inhibit IP_3_R activity in addition to acting as a gap junction blocker [25], we investigate both hypotheses with our model.

## 2. Results

### 2.1. Effect of Different Gap Junction Blockers on ATP-Induced Ca^2+^ Transients in Deiters’ Cells in the Apical and Middle Turns of the Mouse Cochlea

#### 2.1.1. Octanol Had No Effect on ATP-Evoked Ca^2+^ Transients in Deiters’ Cells in Either Cochlear Regions

DCs from hearing mice (P14-P17 for the CBX experiment and P15–19 for the octanol experiment) were investigated in hemicochlea preparation (Figure 1A) after single-cell dye electroporation with Oregon Green BAPTA-1 dye. Cells in the apical (Figure 1B) and middle (Figure 1C) regions were selected for this study. After a baseline record, a control ATP (100 μM) stimulus was administered to the perfusion chamber for 30 s. Both apical and middle cells were activated by the control ATP stimulus. To test the gap junction role in calcium transients of the cells, we administered octanol (1 mM) or CBX (100 µM), non-selective gap junction blockers, to the perfusion for at least 15 min and repeated the ATP stimulation. After the second ATP-induced response, we washed out the gap junction blocker (for at least 15 min) and a third ATP stimulus was administered to test the cell viability and responsiveness (Figure 1D). 

Octanol did not modify the ATP-induced responses compared to control responses (Figure 2). Neither the amplitude (*p* = 0.233, Figure 2B,F), the duration (time between the 50% of the uprising and decreasing part of the transient) (*p* = 0.299, Figure 2C,G) nor the area under curves (*p* = 0.261, Figure 2D,H) of the transients were different tested by the linear model. The age of the experimental animals started at P15 and ended at P19 in both regions (Appendix A). Table 1 and Figure 2 present the pooled data according to age.

#### 2.1.2. Carbenoxolone Significantly Decreased the ATP-Induced Ca^2+^ Transients in Deiters’ Cells in Both Cochlear Regions

We performed the same experimental protocol as the one previously described, in which octanol was used to test the CBX effect on the ATP-induced Ca^2+^ transients of DCs from hearing, P14–17 mice (Figure 1). After 15 min of incubation with 100 µM CBX, the 100 µM ATP was able to elevate the intensity of the dyed cells only slightly (Figure 3, Appendix A). Compared to the control ATP response, amplitude (*p* < 0.001, Figure 3B,F), duration (*p* < 0.05, Figure 3C,G) and AUC values (*p* < 0.001, Figure 3D,H) were significantly decreased (tested by the linear model after logarithmic transformation).

#### 2.1.3. Tonotopic Differences Were Found in Ca^2+^ Transients under Treatment with CBX, but Not in Octanol-Treated Cells

The hemicochlea preparation provides the advantage of studying and comparing cells in the organ of Corti at different frequency regions of the cochlea. We have compared the ATP responses and the effect of gap junction blockade on them in DCs of the apical vs. middle locations, two cochlear regions sensing the most relevant frequency range in mice (Figure 1, Figure 2, Figure 3 and Figure 4). ATP evoked a Ca^2+^ transient in the apical region with a smaller amplitude in the CBX set of experiments, and that resulted in a significant difference between the regions (ages of the experimental mice: P15–P17, Figure 4D, Appendix A). On the other hand, the duration of the responses seemed tendentiously smaller in the middle turn (Figure 4E,F), with a similar AUC in the two regions (Figure 4).

We also detected a visible, but non-significant, difference between the control groups’ octanol and CBX-treated cells (Figure 4) in the apical region: the linear model was fitted to the control groups, age of the animal and region of the cell. Age was included in the linear model because the median age of the CBX-treated animals was 1.5 days less. This difference could have influenced the results because of the post-natal hearing development [2]. The *p*-values for the amplitude, duration, and AUC, respectively: for the groups: 0.23, 0.38 and 0.6; for the age of the animal: 0.93, 0.6 and 0.89; for the region of the cell: 0.04, 0.92 and 0.42. As they are not statistically different and the amount of overlapping data points is considerable (Appendix A), we could treat them as one control group, but to be more precise and gain more information, we handled them separately during the mathematical modelling process and discussed the possible differences.

### 2.2. Mathematical Modelling of Ca^2+^ Transients

#### 2.2.1. The Model

For modelling Ca^2+^ transients and gap junction coupling of DCs in hearing *BALB*/*c* mice, we adapted and merged models from Taheri et al., De Pittá et al. and Wu et al. (Figure 5) [29,30,31,32]. Our model consisted of a system of three differential equations: intracellular Ca^2+^ concentration, IP_3_ concentration, and the activated fraction of IP_3_ receptors (h). It describes IP_3_ dynamics by including in the model the IP_3_ production by agonist-dependent (PLC_β_) and independent (PLC_δ_) enzymes, the IP_3_ degradation by IP_3_-3-kinases and phosphatases, and the IP_3_ diffusion through gap junctions. Ca^2+^ elevation results from Ca^2+^ influx from the endoplamic reticulum (ER) through IP_3_ receptors activated by the IP_3_ elevation and Ca^2+^ influx through P2X receptors activated by ATP. Plasma membrane Ca^2+^-ATPase (PMCA) and sarco-endoplasmic Ca^2+^-ATPase (SERCA) are decreasing free Ca^2+^ concentration (Figure 6). ATP was modified dynamically in the same way as in the experimental chamber.

The model included 28 parameters (Table 1), and all in all, we tested 453,668 parameter combinations. With our model, we reliably simulated calcium transients in the apical and middle regions (Figure 6). The most reliable parameter combinations for the apical and middle regions were slightly different from each other. In the apical region (Figure 6A), we found overlapping parameters between octanol- and CBX-treated cell controls. However, in the middle region (Figure 6B), there were no overlapping combinations, and for this reason, we then analysed the parameter combinations separately. In Table 2, the median ± standard deviation of the most reliable parameters is presented.

In addition to the control Ca^2+^ transients, we simulated the effect of gap junction blockers. For this, we removed the model equations which represent IP_3_ diffusion through gap junctions. For CBX-treated cells of the apical region, no set of parameters was found for which the model predicted the experimental data (Figure 7A). In the middle region, 801 parameter sets fitted well enough to accept them (Figure 7C). As octanol did not significantly affect the transients, we found fitting parameters more easily, and 129,947 and 156,849 sets were found in the apical (Figure 7A) and middle regions (Figure 7C), respectively.

We also tested the hypothesis about the IP_3_ dynamics blocking effect of CBX by removing the IP_3_ activation part (next to the gap junction part). We obtained 501 and 6243 parameter sets for CBX-treated apical (Figure 7B) and middle (Figure 7D) cell transients, respectively. For octanol-treated cells, a great number of parameter sets were found (1642 in the apical (Figure 7B) and 244,933 in the middle (Figure 7D) region).

#### 2.2.2. Analysis of the Parameter Combinations

All transients were characterised by seven points: the time and value coordinates of the 5%, 10%, 50%, 90% and 100% of the intensity on the rising part of the transient and 90% and 50% of the intensity on the decreasing part of the transient. A parameter combination was accepted when the generated curve fitted to experimental data for at least three time points and three value points tested by t-tests. More parameter combinations could reproduce similar curves. We analysed all accepted combinations to outline possible differences in intracellular mechanisms.

Initial values of the intracellular free Ca^2+^ concentration, IP_3_ receptor gating variable, and the intracellular IP_3_ concentration (c_i_, h and IP_3_) have a uniform distribution in the accepted combinations and in the tested parameter space, which means that they would not limit the model success (Appendix A). However, other parameters seem to be more limiting factors.

Parameters reflecting Ca^2+^ transport across the plasma membrane: the rate of the Ca^2+^ influx through the P2X receptor (r_P2X_), and the PMCA parameters (K_PMCA_ and v_PMCA_) showed that in controls of CBX-treated apical cells, lower influx rates are more prevalent than in other control groups, and K_PMCA_ remained lower in the same group (2.1–2.4 µM) in 75% of accepted parameters. In both control groups, cells from the middle regions tend to obtain higher values (2.4–2.9 µM in 58% for controls of octanol- and in 88% for controls of CBX-treated cells). V_PMCA_ showed great variability, and no clear tendency has been seen in either group (Figure 8).

The parameters connected to the ER membrane activity are the rate of the Ca^2+^-induced Ca^2+^-release (r_C_) and the SERCA parameters (v_ER_ and K_ER_) (Figure 9). The r_C_ was tested from 1 to 10, but the accepted values are from 1 to 7. For cells of the apical region, the most prevalent accepted values were 5, while they varied for cells of the middle region: in the case of the control of octanol-treated cells, values from 3 to 5/s represented 96% of all accepted, however in the control of CBX-treated cells, these values represented only 43% of all, and both lower and higher values were accepted. The parameter V_ER_ varied more, but the controls of the CBX-treated cells have the higher accepted values—perhaps in compensation for the greater influx of the calcium-induced calcium release (CICR). For the same reason, K_ER_ took the lowest values in this group (38% were between 0.05 and 0.08). In other groups, 0.1 and higher values were predominant.

The detailed IP_3_ dynamics and the enzymes which are connected to the production and degradation of IP_3_ molecules were also investigated and analysed (Figure 10). During the IP_3_ production, PLC_β_ and PLC_δ_ enzymes are active. PLC_β_ is activated by the agonist, and the parameter (v_β_) was tested from 0.1 to 0.5. In most groups, nearly 50%–50% were the lower and higher parameter distributions, except in the CBX-treated control group, where 99% of the accepted values are higher than 0.3. PLC_δ_ is activated by the higher cytoplasmic Ca^2+^ concentration, and their activation was presented by three parameters: K_PLCδ_, κ_δ_, v_δ_. In the apical region, more than 90% of K_PLCδ_ values are higher (from 0.1 to 0.15), and this is more prominent in the middle-located controls of octanol-treated cells where most (90%) values are over 0.13. The parameter K_δ_ in the middle region took lower values than detected in the apical region: for the control of octanol cells, the apical region parameters are less than or equal to 1.3 in 34%, while in 94% for the middle region. In the control of CBX-treated cells, 17% of the accepted parameters were within the same range of values for apical cells while 35% for middle cells. In contrast, the v_δ_ parameter in the apical region had higher values than in the middle region. In the apically located cells, values were greater than or equal to 0.03 in 58% and 84%, while for cells of the middle region, they were 60% and 58% in the octanol and CBX controls, respectively.

Inositol-triphosphate-kinases (3K) and 5-phosphatases are involved in the degradation of IP_3_ molecules. For the 3K, three parameters were introduced to the system: K_D_, K_3_ and v_3K_. For K_D_, we mostly found lower values in the accepted parameter combinations. In all cases, more than 60% of the parameters were less than or equal to 0.05, and were even more predominant in the controls of octanol-treated cells (93% in both regions).

The K_3_ parameter is the IP_3_ affinity of IP_3_-3K. For controls of octanol-treated cells, K_3_ had higher values (63% and 95% in the apical and middle regions, respectively. Higher or equal to 1). The values for the controls of CBX-treated cells were similar in the apical region (60%), but in the middle region, mostly lower values were accepted (only 30% were higher or equal to 1).

The v_3K_ parameter, the maximal degradation rate of the kinases, was higher in the apical region compared to the middle region (38% and 8% compared to the 58% and 27% in the controls of octanol- and the CBX-treated cells, respectively, were higher than 3).

The 5-phosphatases, the other degrading enzyme family, were represented by the r_5P_ parameter. The degradation rate in the apical region was slower: 63% and 93% of all the accepted values were equal or lower than 0.05 compared to 45% and 12% in the middle region (in the controls of octanol- and CBX-treated cells, respectively).

When comparing the accepted parameters between treated cells and controls, the initial values remain of the same distribution (Appendix A), emphasising that they are not limiting the model. All in all, removing the gap junction part of the equation did not significantly change the distribution of the parameters in either region (Appendix A). However, parameters for the Ca^2+^ transport through the plasma membrane were visibly different: the rate through P2X receptors decreased, and was even more prominent for the CBX treatment while the V_PMCA_ parameter of the apical region increased (Appendix A). The other pump velocity V_ER_ was also elevated in the same group. The affinity parameters (K_PMCA_ and K_ER_) were differently changed: parameters for the plasma membrane were increased while parameters for the ER membrane were decreased, but only in CBX-treated cells (Appendix A).

By comparing the IP_3_ enzyme dynamics in treated and control groups, we saw that they became limiting parameters (only one or two values were accepted) of the model after removing the equations representing the fluxes through gap junction channels and IP_3_ receptors (Appendix A). K_PLCδ_ in CBX-treated cells of the apical region is 0.09 µM, and in most octanol-treated cells is 0.07 µM. The inhibition constants, κ_δ_, and the maximal rate of the production, v_δ_, were similarly limiting these groups: in most octanol-treated cells, κ_δ_ and v_δ_ are 1.7 µM and 0.03 µM·s^−1^, respectively. In CBX-treated cells, the values were 1.5 and 2 for κ_δ_, and 0.02 and 0.05 for v_δ_. Interestingly, in the middle region, all of these parameters varied considerably in this group; however, the model in which only the gap junctional part was removed became limited, and the values were mostly 0.09 µM for K_PLCδ_, 1.5 µM for κ_δ_ and 0.05 µM·s^−1^ for v_δ_ (Appendix A) in CBX-treated cells only.

For the IP_3_ degradation, the 3-kinase activation parameters showed similar trends (Appendix A). In the apical region, only one or two parameter values were accepted in the case of gap junction- and IP_3_-removed models: the Ca^2+^ affinity of these enzymes (K_D_) is 0.7 µM (for octanol-treated cells), and 0.3 and 1 µM (for CBX-treated cells). The IP_3_ affinity, K_3_, was 0.7 µM, and 0.0.9 and 1.2 µM for the octanol- and CBX-treated cells, respectively, while the maximal rate was 1 µM·s^−1^ for the octanol-treated cells, and 3 and 4 µM·s^−1^ for the CBX-treated cells. Again, these groups showed great variability in the middle region, but for the CBX-treated cells, gap junction equation removed groups were limited: 1 µM for the K_D_, 0.9 µM and 3 µM·s^−1^ for the K_3_ and v_3K_, respectively.

The last enzyme group involved in the IP_3_ degradation, the 5-phosphatases, had similar limitations (Appendix A). The rate of these enzymes in the apical region, gap junction and IP_3_ equation removed groups was 0.02 s^−1^ for octanol-treated cells, and 0.03 and 0.09 s^−1^ for CBX-treated cells. In the middle region, only the CBX-treated and gap junction removed group showed limitation: most of the accepted values were 0.09 s^−1^.

## 3. Discussion

Both ionotropic P2X and metabotropic P2Y purinergic receptors are present in cochlear supporting cells [10,18,19,33]. Indeed, ATP induced reversible and repeatable Ca^2+^ transients in DCs in *BALB*/*c* mice in the third post-natal week. Differences in Ca^2+^ transients were observed between apical and middle cochlear regions in the CBX set of experiments. The amplitude of Ca^2+^ responses was larger, while their duration was shorter in the middle turn. Other studies showed that the distribution of P2Y and P2X receptors and their subtypes depends on the tonotopic location of the DCs and the age of the animals [2].

The inbred *BALB*/*c* mouse strain shows age-related hearing loss and is also used as a presbycusis model [34,35]. Detailed anatomy of the organ of Corti and their cells vary in different mice strains, most prominently in the apical region [36]. The volume of the cells could influence the Ca^2+^ handling mechanism, which may indicate that our results mostly describe the responses of the Deiters’ cells in the *BALB*/*c* mice strain.

Our animals aged P15–19 for the octanol- and P14–17 for the CBX-treated experiments, with similar distribution between the two cochlear regions. We did not expect to find differences between this short interval difference; however, studies showed that the different types of purinergic receptors expression could change over post-natal development [2,37,38,39]. The age difference between the animals (two days difference) may explain the non-significant, but visible difference between the controls of the two groups. Because of this difference, we analysed the control groups separately.

A higher amplitude in the middle region could imply a stronger and faster response to the stimulus, perhaps due to a difference in the number and type of purinergic receptors, PLC or IP_3_R, or a greater endoplasmic Ca^2+^ store. Unfortunately, most of these components have not been investigated in the cochlea before. Tonotopic differences were found (or hypothesised) both on supporting cells [2,40] and hair cells [15,40,41]. This is supported by our experimental and modelling results, as several components of the Ca^2+^ signalling cascade showed tonotopic differences.

On the other hand, a shorter duration of the response would imply stronger and faster Ca^2+^-removing mechanisms. This could be related to an increased number of SERCA or PMCA pumps. An interesting observation has been made on the ability of OHCs to handle Ca^2+^ load along the tonotopic axis of the cochlea. Two main differences between hair cells of low- and high-frequency regions (apical and basal regions, respectively) have been demonstrated: larger amplitude of the MET current and fewer PMCA pumps in high-frequency OHCs. Both are likely to explain the Ca^2+^ overload in OHCs of the basal turn [15]. One would expect a similar tonotopic gradient in DCs.

Apart from the number of SERCA and PMCA pumps, their isotypes could also be different. The whole inner ear and cell-specific sequencing studies in young (P0) mice have shown that two types of isoforms (ATP2A1 and ATP2A) are present in the organ of Corti, and both are also present in DCs [42,43,44]. However, in older (P2) mice, none of them were found in the cochlea, which could indicate their age-dependent role in hearing [45].

Enzymes which play a role in IP_3_ production and degeneration have also been found in the hearing organ via mRNA sequencing methods in young mice. We modelled the PLCδ and β, which are activated differently. Their isotypes are also found in DCs: PLCδ1, 3 and 4 and PLCβ 1, 2, 3, 4 were detected [42,43,44]. From this, the PLCδ1,3 and PLCβ3,4 have relatively high expression in DCs [44]. The same study showed the presence of inositol-3-kinase-B and C in this type of supporting cells in addition to several different inositol-5-phosphatases.

In addition to enzymes, pumps and cell surface receptors, the IP_3_ receptor also has three different isoforms, and all three are present in the young mouse DCs—but the IP_3_ receptor-2 has the highest expression rate [44]. Our model successfully described all these Ca^2+^ signalling mechanisms.

Cochlear supporting cells are coupled with gap junction channels that allow the propagation of intercellular Ca^2+^ signals via the diffusion of Ca^2+^-mobilizing second messengers [23]. Recent studies have successfully blocked gap junctions in DCs using both 1 mM octanol and 100 μM CBX as gap junction blockers [19,46]. Interestingly, our experimental results showed that 1 mM octanol did not have any effect on ATP-evoked calcium responses in DCs (Figure 2). This could be related to a decreased expression of Cx26 and Cx30 through ageing, which could impair the effect of octanol on blocking gap junction channels. Indeed, age-related hearing loss might be associated with changes in the expression patterns of connexins in cochlear cells [47]. Although, no research has demonstrated such connexin expression decrease in DCs of 15–19 days old mice. In addition, Ca^2+^ waves in supporting cells of P25-P35 mice were inhibited by 1 mM octanol, and thus it is unlikely that connexin expression is already decreasing in P14-P19 mice [19]. Similarly, 10 mM octanol was two-fold less effective in blocking hemichannel-mediated IP_3_ release in the cochlear sensory epithelium of adult guinea pigs compared to 100 μM CBX [48]. As mentioned earlier, cochlear gap junctions are mainly composed of Cx26 and Cx30 [23]. Octanol is usually referred to as a non-selective gap junction blocker; however, some studies mentioned that it is selective for Cx43, Cx46 and Cx50 isoforms [25,49]. Altogether, these questions the effectiveness of octanol in blocking gap junction channels in cochlear cells.

CBX can also be used to study gap junctional coupling. It is one of the most widely used gap junction blockers [48]. A 100 μM CBX has previously been used to block gap junctions in DCs [19,48]. Our Ca^2+^ imaging experiment showed that CBX had a stronger effect on the middle region, suggesting that more connexins are present in the middle part of the cochlea than in the apical one. In the absence of CBX, the amplitude of the calcium responses was higher in the middle region, with a P2/P1 ratio being smaller. The P2/P1 ratio emphasises how CBX treatment influenced Ca^2+^ responses in each cochlear region (Figure 3,4). The expression gradient of connexins was found to decrease from base to apex in inner and outer sulcus cells in 5 to 6 days old mice [23]. We could suggest that DCs are also following the same tonotopic pattern. However, Zhao et Yu found that the expression of Cx26 and Cx30 in supporting cells was higher in the apical region than in the basal region in adult guinea pigs [50]. Unless it is a species-specific difference, the latter supports the hypothesis of the blocking effect of CBX on IP_3_R activity, in addition to acting as a connexin blocker [25]. In this case, its stronger effect on cells of the middle region could be explained by its specificity for other targets outside of gap junctions. Overall, we need further evidence of gradient distributions of connexin expression in mice cochlear cells, especially on DCs, and we need to further investigate the mechanisms of action of CBX.

In our modelling results, parameter analysis also showed some side effects of the blockers, which could help future studies of these drug effects in living cells.

Building and analysing the first Ca^2+^ model for cochlear DCs could be relevant to understanding both the tonotopic differences and developmental changes in the hearing organ.

## 4. Materials and Methods

### 4.1. Collection of Experimental Data from Ca^2+^ Imaging of DCs

All the experimental procedures and the animal care were in accordance with the National Institute of Health Guide for the Care and Use of Laboratory Animals. The procedures were approved by the Animal Use Committee of Semmelweis University, Budapest.

*BALB*/*c* strain mice of both sexes were sacrificed between post-natal day 14 (P14) to P19. The apical regions P15–17 were used for CBX experiments and P15–19 for the octanol experiment. In the middle region, P14–17 mice were sacrificed for the CBX experiments and P15–19 for the octanol experiments. The experiment protocol was based on previous studies [1,6,8]. The mice were decapitated under isoflurane anaesthesia, and their cochleae were removed and placed in an ice-cold oxygenated solution similar to the perilymph (composition in mM: NaCl 22.5, KCl 3.5, CaCl2 1, MgCl2 1, Hepes 10, Na-gluconate 120, glucose 5.55; pH 7.4; 320 mOsm/L). The cochleae were placed into the cutting chamber of a vibratome (Vibratome Series 3000, Technical Products International Inc., St. Louis, MO, USA) and halved along the midmodiolar plane while continuously bathed in the perilymph-like solution. Hemicochlea were placed into an imaging chamber filled and perfused (speed: 3.5 mL/min) with the oxygenated experimental solution. The cells of interest were selected under a LUMPlanFl 40x/0.80w water immersion objective (Olympus, Tokyo, Japan) with oblique illumination, and were loaded by single-cell electroporation. For this procedure, the borosilicate pipettes (5–7 MΩ; Harvard Apparatus, Holliston, MA, USA) were filled with light excitable Ca^2+^ indicator Oregon Green 488 BAPTA-1 hexapotassium salt (OGB-1) (ThermoFisher Scientific, Waltham, MA, USA) dissolved in distilled water (final concentration of 1 mM). The pipettes were mounted onto an electrode holder, which was attached to a micromanipulator (Burleigh PCS-5000, Thorlabs, Munich, Germany). As the pipettes approached the cells of interest, a single square wave current impulse (10 ms, 10 μA) was applied, allowing the cells to be loaded with OGB-1. The current was generated by a pCLAMP10 software-guided stimulator system (Biostim STE-7c, Supertech Ltd., Pecs, Hungary; MultiClamp 700B Amplifier and Digidata 1322A, Molecular Devices, Budapest, Hungary).

The calcium imaging procedure was performed at room temperature. Excitation light of 494 ± 5 nm (Polychrome II monochromator, TILL Photonics, Planegg, Germany) was used to illuminate the cells, and the light emitted was recorded after passing through a band-pass filter (523 ± 25 nm). An Olympus BX50WI fluorescence microscope (Olympus, Tokyo, Japan) equipped with a Photometrics Quantix cooled CCD camera (Photometrics, Tucson, AZ, USA) was used and was controlled with the Imaging Workbench 6.0 software (INDEC BioSystems, Los Altos, CA, USA). The images were taken at a frame rate of 1 Hz during the drug-evoked responses, and the rest of the time, the frame rate was 0.1 Hz to reduce phototoxicity and photobleaching. In the perfusion chamber, the volume of the buffer was about 1.9 mL. The cells were stimulated by 100 µM ATP (Sigma–Aldrich, St. Louis, MO, USA), and added to the perfusion for 30 s. Before ATP stimulation, a baseline period of at least 3 min was registered.

For the investigation of the gap junctional coupling mechanism, we perfused the cells with gap junction blockers: octanol (1 mM, stock solution in ethanol) and carbenoxolone disodium salt (CBX; 100 μM, stock solution in distilled water; both drugs from Sigma-Aldrich, St. Louis, MO, USA). After the 1st ATP stimulus, the respective blocker was added to the perfusion for at least 15 min before the 2nd ATP administration. After the ATP response had been completed, the blocker-containing solution was replaced with a perilymph-like solution for at least 15 min before exciting the cells with a 3rd ATP dose (Figure 1D). To exclude the effect of the octanol solvent alcohol, we repeated the experiments with the respective concentration of ethanol (n = 3). No effect was detected on either the baseline or the ATP response.

We have performed our experiments on DCs in the apical and middle turns in the hemicochlea preparation. These turns include the most relevant frequency region of the basal membrane (~0.5–30 kHz; [35,51]), and the success rate of perfect basal region cut during the hemicochlea preparation is low.

Fluorescence intensities were background-corrected using a nearby area devoid of loaded cells. The relative fluorescence changes were calculated as follows: dF/F_0_ = (F_t_ − F_0_)/F_0_, where F0 is the fluorescence intensity of the baseline, and Ft is the fluorescence intensity at time t [1]. A total of 7 characteristic point coordinates (time and dF/F_0_ values) were recorded: 100% (maximal amplitude), 5%, 10%, 50% and 90% of the maximal amplitude on the increasing part of the transients, and the 90% and 50% of the maximal amplitude on the recovery part of the transients. The duration of the transients was calculated as the time between 50% of the maximal amplitude at the increasing and descending part of the response.

Statistical analysis of the experimental results was performed in R 4.1.0. version. The data distribution was tested by Shapiro–Wilk test. Most of the data were not normally distributed; because of this, these were logarithmically transformed, and a linear regression model was used.

### 4.2. Mathematical Modelling and Comparison with Experimental Data

Previous studies have highlighted similarities between auditory supporting cells and glial cells [5]. Therefore, our model is based on studies in which calcium dynamics in astrocytes were simulated [13,14]. Taheri et al. [13] and De Pittà et al. [14] models account for Ca^2+^ dynamics evoked by neurotransmitters. A gap junctional flux of IP_3_ was added to our model employing Wu et al. [15] model.

The model is an open-cell model in which the intracellular Ca^2+^ concentration can change due to exchange across the plasma membrane. We assumed that the spatial arrangement of calcium concentration is uniform in the cytosol and ER. The model consists of a system of three differential equations: the changes in cytosolic Ca^2+^ concentration (1), the activated fraction of IP_3_R (2) and the IP_3_ intracellular concentration (3):(1)$$\frac{dc}{dt}=J_{P2X}\;–\;J_{PMCA}+J_{IP3R}\;–\;J_{SERCA}+J_{leak}\;,$$
(2)$$\frac{dh}{dt}=\frac{h_{\infty }-h}{\tau },$$
(3)$$\frac{d\left[IP_{3}\right]}{dt}=v_{ATP}+v_{\delta }\;–\;v_{3K}\;–\;v_{5P}+{4\times J}_{GJC\;}.$$

The *J_i_* represents the fluxes through pumps and channels embedded in the plasma membrane and ER membrane, respectively (Figure 5). *J_P2X_* represents the influx of Ca^2+^ to the cytosol through P2X channels:(4)$$J_{P2X}=r_{P2X}\;\left(1-c\right)\;\left[ATP\right].$$

*J_PMCA_* is the outflux of Ca^2+^ across the plasma membrane through PMCA pumps. *J_SERCA_* is the flux of Ca^2+^ from the cytosol into the ER through SERCA pumps:(5)$$J_{PMCA}=v_{PCMA}\frac{c^{2}}{\left(c^{2}+K_{PMCA}^{2}\right)}\;$$

And
(6)$$J_{SERCA}=v_{ER}\frac{c^{2}}{\left(c^{2}+K_{ER}^{2}\right)}.\;$$

*J_IP3R_* is the flux of Ca^2+^ from the ER to the cytosol through the IP_3_R channel. This model simplifies the opening probability of IP_3_R by assuming that the gating kinetics is mainly driven by three binding sites on the receptor: an IP_3_ binding site (*m*), a Ca^2+^ binding site for activation (*n*), and a Ca^2+^ binding site for deactivation (*h*). The dynamics of *h* are governed by the equation given previously, where *J_IP3R_* is given by:(7)$$J_{IP3R}=\left(r_{release}\;c_{1}\;m_{\infty }^{3}n_{\infty }^{3}h^{3}\right)\left(c_{ER}-c\right).\;$$

With
$$c_{ER}=\frac{c_{0}-c}{c_{1}},m_{\infty }=\frac{\left[IP_{3}\right]}{[IP_{3}]+d_{1}},n_{\infty }=\frac{c}{c+{\;d}_{5}},h_{\infty }=\frac{Q_{2}}{Q_{2}+c},\tau =\frac{1}{a_{2}\;(Q_{2}+c)},\;\mathrm{and}\;Q_{2}=d_{2}\frac{\left[IP_{3}\right]+{\;d}_{1}}{[IP_{3}]+d_{3}}.$$

*J_leak_* is a nonspecific calcium leakage from the ER:(8)$$J_{leak}=\left(r_{leak}\;c_{1}\right)\left(c_{ER}-c\right).\;$$

IP_3_ is produced from the hydrolysis of PIP_2_ catalysed by two PLC isoenzymes, PLC_δ_ and PLC_β_. The binding of ATP to G-protein coupled P2Y receptors induces the activation of PLC_β,_ releasing IP_3_ (*v_ATP_*). PLC_δ_ is mainly activated by intracellular Ca^2+^ elevation (*v_δ_*). IP_3_ concentration depends on its production and its degradation via two major pathways. The first degradation pathway is through the dephosphorylation of IP_3_ by the inositol polyphosphate 5-phosphatase (IP-5P) (*v*_5*P*_). The second one is through Ca^2+^-dependent phosphorylation of IP_3_ by the IP_3_ 3-kinase (IP_3_-3K) (*v*_3*K*_). Both reactions can be considered as Michaelis–Menten type. However, IP-5P is not likely to be saturated by *IP*_3,_ so the rate of IP_3_ degradation can be linearly approximated. The equations are:(9)$$v_{\delta }=\frac{\bar{v_{\delta }}}{1+\frac{\left[IP_{3}\right]}{\kappa_{\delta }}}\;\frac{c^{2}}{c^{2}+K_{PLC\delta }},\;$$
(10)$$v_{\delta }=\frac{\bar{v_{\delta }}}{1+\frac{\left[IP_{3}\right]}{\kappa_{\delta }}}\;\frac{c^{2}}{c^{2}+K_{PLC\delta }},\;$$
(11)$$v_{3K}=\bar{v_{3K}}\;\frac{c^{4}}{c^{4}+K_{D}^{4}}\;\frac{\left[IP_{3}\right]}{\left[IP_{3}\right]+K_{3}},\;$$
(12)$$v_{5P}=\bar{r_{5P}}\;\left[IP_{3}\right].\;$$

In addition, we incorporated a gap junctional flux (*J_GJC_*) into our model, taking into consideration that IP_3_ can diffuse through gap junctions. To this end, we used Wu et al. [15] model, in which a system of *i* cells is considered. To simplify this model, we assumed the IP_3_ concentration of the neighbouring cells constant over time, at a concentration of 5 μM, and IP_3_ diffusing only from the neighbouring cells to the target Deiters’ cell. Thus, the diffusion rate of IP_3_ only depends on its production and degradation over time in the experimental target cell. In Equation (13), we multiplied *J_GJC_* by 4 because DCs usually have four neighbouring cells, as follows:(13)$$J_{GJC}=\gamma \;\left(5-\left[IP_{3}\right]\right).\;$$

We opted to treat ATP as inputs to our model (14). In our experiment, DCs were stimulated by injecting ATP for 30 s in the perfusion chamber. We assumed that ATP exponentially saturates to a certain level (*s_∞_*), and then exponentially decays before becoming depleted.
(14)$$ATP\left(t\right)\;\left\{\begin{array}{c} 0\;\;\;\;\;\;\;\;\;\;\;\;\;\;\;\;\;\;\;t<t^{*}\\ s_{\infty }\;\left(1-e^{-r_{rise}\;\left(t-t^{*}\right)}\right)\;\;\;\;\;\;\;\;\;\;\;t^{*}\leq t<t^{*}+d_{rise}\\ A\;e^{-r_{decay}\;\left(t-\left[t^{*}+d_{rise}\right]\right)}\;\;\;\;\;\;\;t^{*}+d_{rise}\leq t \end{array}\right.\;$$
where
$$s_{\infty }=\frac{A}{1-e^{-r_{rise}{\;d}_{rise}}},\;\mathrm{and}\;r_{decay}=-\frac{1}{d_{decay}}\log(\frac{0.005}{A}).$$

*t** represents the starting time of ATP stimulus, *A* is the maximum amplitude, *r_rise_* and *r_decay_* represent the rates of rise and decay, respectively, and *d_rise_* and *d_decay_* are the durations of the rising and decaying phases of ATP concentration. We estimated the ATP concentration in the perfusion chamber based on the velocity of the solution in which ATP was injected (0.058 mL.s^−1^), the volume of the chamber (2 mL), and the concentration of the injected ATP dose (100 μM). The estimated ATP values in the perfusion chamber were compared to the ATP model (14), and an estimation of the model parameter values was obtained using a non-linear least squares approach (function nls in R). The initial set of ATP model parameters were: Maximal amplitude (A) = 58.85 μM; *d_rise_* = 30 s. *r* was fitted first, and *d_decay_* second, both using a non-linear least squares approach. The following values were obtained and used for the model: *r_rise_* = 0.0296 s^−1^, *d_decay_* = 316.7 s.

We implemented our model in R 4.1.0 using forcing functions of the deSolve package with the aim of adjusting the parameter values to fit the model to our experimental data.

Model parameters were tested in a parameter space by random sampling (Table 3).

From each result of the parameter combinations, the same characteristic points were recorded as in the case of the experimental data: 100% (maximal amplitude), 5%, 10%, 50% and 90% of the maximal amplitude on the increasing part of the transients, and the 90% and 50% of the maximal amplitude on the decreasing part of the transients. Then, these points were compared to experimental data using *t*-tests. If at least 3 time points and 3 value points showed a non-significant difference from the experimental data, the parameter combination was saved.

## 5. Conclusions

Our experimental results showed that the ATP-evoked calcium transients in DCs in the apical turn were smaller (amplitude, duration and AUC, too) at an earlier developmental stage (P14–17 vs. P15–19), which may represent the correlation of the development of purinergic Ca^2+^ signalling with the mammalian cochlear development that generally proceeds from the basal region to the apex [52,53,54].

The larger ATP response in DCs in the middle turn and in the more mature (P15–19) apical turn may imply a higher number or different subtype pattern of purinergic receptors, PLC, IP3R or a larger endoplasmic Ca^2+^ store. The shorter duration of these calcium signals may be the result of a faster recovery caused by a higher number of SERCA and/or PMCA pumps, eliminating intracellular Ca^2+^.

DCs are coupled with gap junction channels, which influence the Ca^2+^ signalling in these cells. Despite this, the use of 1 mM octanol did not have any effect on the ATP-induced Ca^2+^ transients. Theoretically, this may be linked to a maturation-dependent decrease in the expression of cochlear gap junction proteins Cx26 and Cx30, although this has not been described in P15-P19 mice yet. However, octanol is thought to be selective for the Cx43, Cx46 and Cx50 isoforms which raises a question of its efficiency in blocking gap junction channels of cochlear supporting cells. On the contrary, CBX inhibited the ATP response in both cochlear turns, more so in the middle one, which may be explained by the higher number of CBX-sensitive Cx26/30 in that turn. The inhibitory effect of CBX on the IP_3_-mediated Ca^2+^ release, described recently [25], cannot be excluded.

Based on astrocyte models, we successfully simulated the calcium dynamics in DCs. This supports the hypothesis that DCs are glia-like cells. Our model, using the parameters of our experimental results, suggests the tonotopical distribution of IP_3_ degrading enzymes and the similarity of the CICR rate to the one in the astrocyte model.

## Figures and Tables

**Figure 1 ijms-24-11095-f001:**
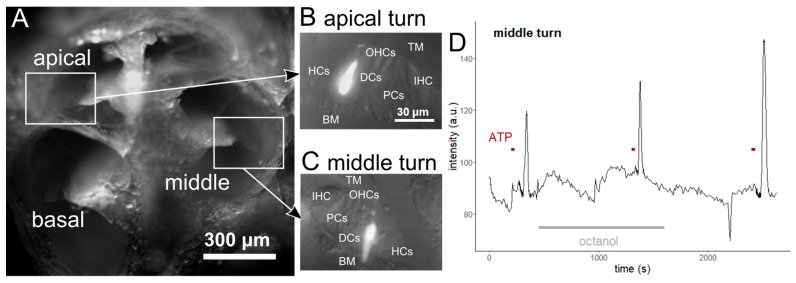
Experimental protocol of the Ca^2+^ imaging experiments. The experiments were carried out on hemicochlea (**A**) of *BALB*/*c* mice with a mature hearing organ (>P14). The hemicochlea preparation allows the investigation of cells in the organ of Corti along the tonotopic axis. Single-cell electroporation was used to load individual DCs with OGB-1 in the apical (**B**) and middle (**C**, same magnification) regions of the cochlea. The good signal-to-noise ratio of the method makes it also suitable for Ca^2+^ imaging in the tiny processes of the DCs. The representative trace (**D**) shows the course of the experiments. Recording of a 3 min baseline intensity (a.u., arbitrary unit) was followed by ATP stimulations (100 μM, 30 s) in the absence (control) and presence of gap junction blocker (octanol or CBX). The blocker was added to the perfusion (3.5 mL/min) at least 15 min before the second ATP application. At the end of the experiments, the gap junction blocker was washed out (for 15 min, at least), and a 3rd ATP stimulus was administered to test the viability and responsiveness of the imaged cell. (TM: tectorial membrane, BM: basal membrane, IHC: inner hair cell, OHCs: outer hair cells, HCs: Hensen’s cells, DCs: Deiters’ cells, PCs: pillar cells).

**Figure 2 ijms-24-11095-f002:**
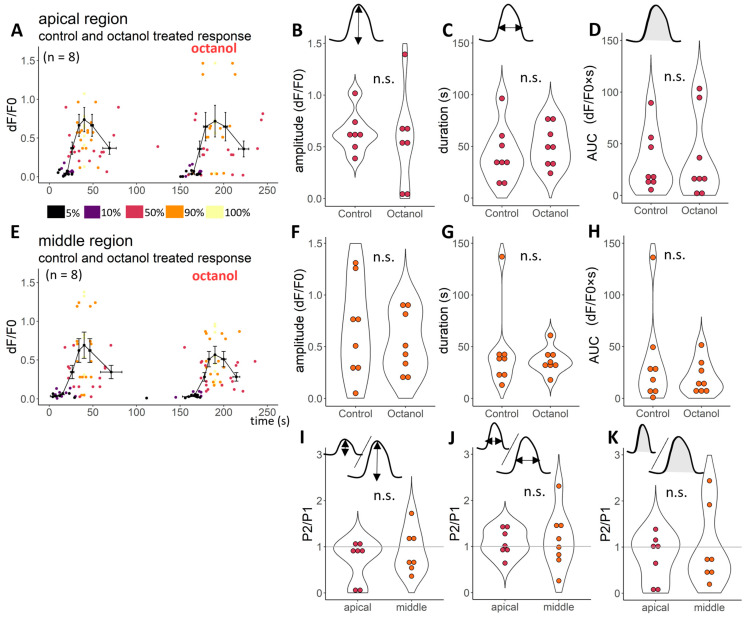
Octanol has no significant effect on ATP-evoked Ca^2+^ transients in DCs in any of the cochlear regions. All the fluorescent transients were characterised with 7 points: on the elevation, we read 5%, 10%, 50%, 90% and 100% of the maximal intensity, whilst on the decreasing part of the transients, 90% and 50% of the maximal intensity were read. There was no difference between ATP responses in the absence and presence (**A**,**E**) of octanol (1 mM) in any of the parameters measured (amplitude (**B**,**F**)—100% of the fluorescent intensity change during the transient, duration (**C**,**G**)—the time difference between reaching and decreasing the 50% of the maximal fluorescent intensity, the area under the curve (AUC, **D**,**H**)) in any of the cochlear regions (upper row: responses of the apical region; below: responses of the middle region). The Ca^2+^ responses were measured in terms of fluorescence changes (dF/F_0_). The average of traces and SEM are from 8-8 tested cells (from P15 to 19 animals) in the apical and middle regions presented on the (**A**,**E**), respectively. The amplitude of one cell (100%) is removed from (**A**,**B**) to visualise better and make it comparable to Figure 3 (average and s.e.m. values are calculated to include this data, too), removed values: 1.7 and 1.54 dF/F0 for control and octanol treated response, respectively. The treatment effect could be visualised by the treatment response/control response ratio (P2/P1). Octanol treatment had no effect on any measured characteristics (P2/P1~1, I, J and K). P2/P1 value (3.59) was removed from panel (**I**), and responses of one apical cell value from P2/P1 of duration and AUC (6.33 and 17) from panel (**J**,**K**), respectively. Significance was tested by the linear model after logarithmic transformation.

**Figure 3 ijms-24-11095-f003:**
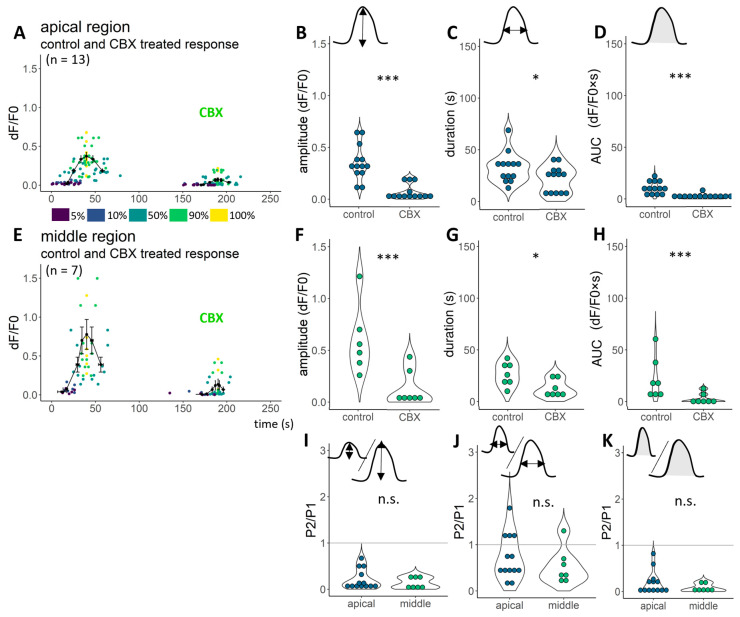
CBX considerably diminishes ATP-induced Ca^2+^ transients. **A** 15 min perfusion with CBX clearly decreased the ATP-induced Ca^2+^ transients both in the apical (**A**) and middle (**E**) regions. All the fluorescent transients were characterised with 7 points: on the elevation, we read 5%, 10%, 50%, 90% and 100% of the maximal intensity, whilst on the decreasing part of the transients, 90% and 50% of the maximal intensity were read. Amplitude (**B**,**F**), duration (**C**,**G**) and AUC values (**D**,**H**) were significantly decreased in the apical and middle regions (respectively). The amplitude of one control cell (100%) is removed from A and B to visualise better and make it comparable to Figure 2 (average and s.e.m. values are calculated to include this data, too), removed values: 1.58 dF/F0. The treatment effect could be visualised by the treatment response/control response ratio (P2/P1; **I**–**K**). CBX treatment had a higher effect on cells of the middle region (**I**,**K**). *** *p* < 0.001, * *p* < 0.05, tested by linear models after logarithmic transformation.

**Figure 4 ijms-24-11095-f004:**
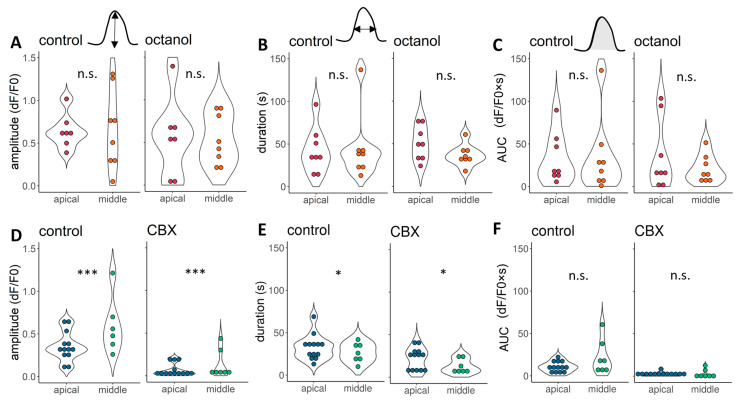
Tonotopic differences were detected between ATP response amplitudes and durations in CBX treatment. **A** 100 μM ATP could activate both apical (deeper colour) and middle (lighter) region DCs. Their amplitudes were smaller in the apical region of the cochlea in the case of the CBX-treated cells (**D**), while no differences between the average were found in the octanol control cells (**A**). Their other response characteristics were different: slightly longer responses were found in the apical region in the case of CBX control cells, which difference remained during the CBX treatment (**E**). The duration was not different between cells from the apical and middle turns either in the octanol control or octanol treated groups (**B**). The AUC values were not different in any of the tested groups (**C**,**F**). All data are pooled and not considering the age of the mice (for this, see: Appendix A). *** *p* < 0.001, * *p* < 0.05, tested by linear models after logarithmic transformation.

**Figure 5 ijms-24-11095-f005:**
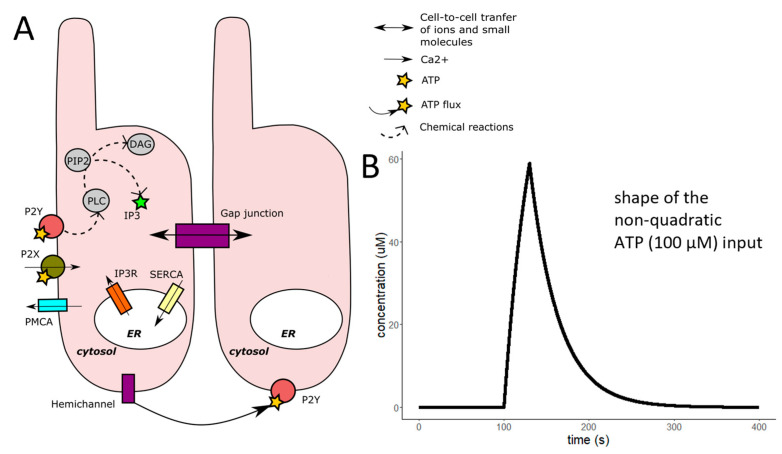
Schematic view of the major biochemical reactions involved in ATP-activated DCs and the effect of gap junction coupling and our non-quadratic ATP input. The model was based on De Pittá et al. 2009 model [31], which simulated extracellular quadratic glutamate activation of glial cells. (**A**) The schematic representation shows the main signaling components and ion and molecular fluxes included in our model. The activation of the metabotropic receptors induces the phospholipase-C (PLC), which produces inositol-trisphosphate (IP_3_) and diacylglycerol (DAG) by the cleavage of inositol-bisphosphate (PIP_2_). IP_3_ then diffuses through the IP_3_ receptors located on the membrane of the internal Ca^2+^ store of the cells and elevates the cytoplasmic Ca^2+^ concentration. IP_3_ is a small molecule, and gap junctions are permeable to them. IP_3_ molecules could be originated from the activated neighbouring cells. To normalise the concentration, the SERCA is activated by the Ca^2+^. We added the ionotropic P2X receptor and the PMCA to the model. The ATP dynamics of our model (**B**) are estimated similarly to the IP_3_ dynamics in Taheri et al., 2017 model [29,30], but we modified it so that it could fit the ATP concentration in our experimental chamber during the ATP stimuli (30 s, 100 μM). An additional gap junction component was added employing Wu et al. 2005 model [32].

**Figure 6 ijms-24-11095-f006:**
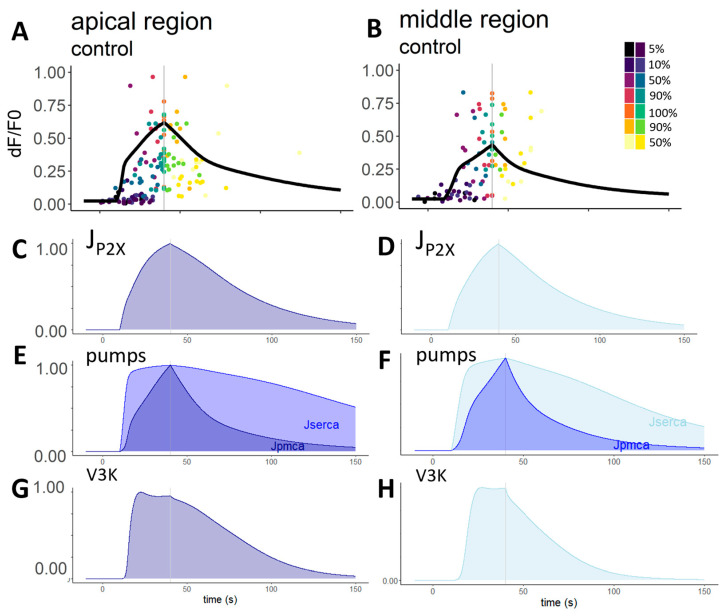
Model results in the apical and middle regions with the median parameters. The panel (**A**,**B**) shows the model curves resulting from the median parameters of the apical and middle regions, respectively, with the experimental data points. From the equations, we could make assumptions about the different intracellular mechanisms and their activation in time (their values are normalised). Influx through the P2X receptors has a very similar shape to that of the Ca^2+^ transient, highlighting its importance in calcium signalling ((**C**) in the apical and (**D**) in the middle regions). The pump mechanism curve shows the expected difference, as PMCA pumps are usually less sensitive and activated by higher Ca^2+^ concentrations than SERCA pumps. They were nearly identical in the case of the apical (**E**) and middle regions (**F**). The inositol-3-kinase (*v*_3*K*_), which plays a role in the degradation of the IP_3_ molecule, has a plateau phase but decreases faster in cells of the middle region (**H**) compared to the apical region (**G**).

**Figure 7 ijms-24-11095-f007:**
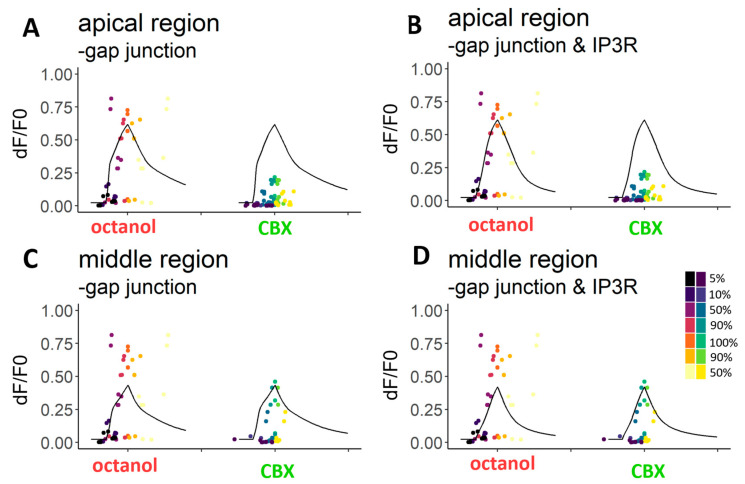
Modelling the effects of gap junction blockers. Removing the gap junction (**A**,**C**) and IP_3_ receptor (**B**,**D**) equations from the model could simulate the responses of octanol-treated cells, but not precisely the responses of CBX-treated cells. The ATP-induced Ca^2+^ transients in the presence of octanol did not significantly differ from the original ATP-induced transients, neither in the apical (**A**) region nor in the middle (**C**) region. However, CBX-treated cells were significantly less sensitive to ATP, and their simulated Ca^2+^ response curves were unreliable when the same parameter values as the control curves were used ((**A**), see Table 2). In the middle region (**C**), more reliable parameters were found. It was hypothesised that CBX blocks the IP_3_ pathway. We removed the equations representing Ca^2+^ efflux through IP_3_R from the model. Responses of both regions and both drugs treated cells could be modelled (**B**,**D**). Parameters were accepted if 3 of the 7 chosen points were reliably fitted. The 7 different points of the transients: 5%, 10%, 50%, 90% and 100% of the rising and 90% and 50% of the decreasing part of the transients.

**Figure 8 ijms-24-11095-f008:**
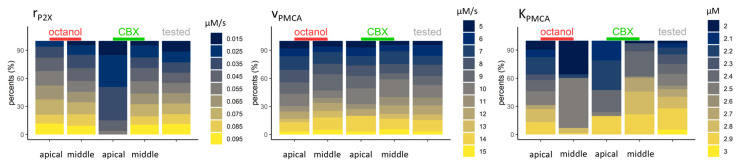
Parameters linked to Ca^2+^ transport across plasma membrane could vary over wide ranges of values in ATP-induced transients of control cells. The parameter of influx rate through the r_P2X_ varied between 0.015 and 0.095 µM·s^−1^. In octanol controls and CBX control in the case of the middle region, this parameter does not seem to be a limiting parameter—all the parameter values are represented similarly as the tested parameters, whereas the control of CBX-treated cells in the apical region has lower values (0.015–0.055 µM·s^−1^). The results of the pump parameter V_PMCA_ varied from 5 to 15 in each group (here, only the rounded values are presented, the original step between the two values is 0.01). The other pump parameter, K_PMCA_, however, differed in all groups. In cells from the apical region, it could take values in the whole interval, whilst in the middle region, it takes higher values in more than half of the cases (2.4–3 µM·s^−1^).

**Figure 9 ijms-24-11095-f009:**
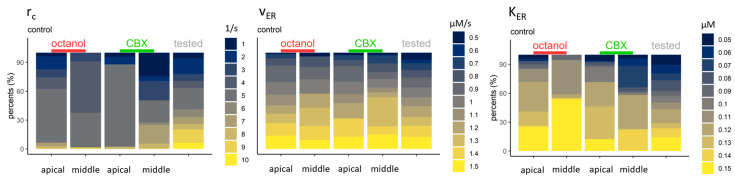
Parameters connected to the Ca^2+^ transport through the ER membrane. The accepted rate of the r_C_ could highly limit the success of the model. It is mostly between 1 and 5, except for the controls of the CBX-treated cells in the middle region, where 7 takes nearly 20% of the accepted values. To compensate for the increased Ca^2+^ influx into the cytoplasm, VER (maximal velocity of the SERCA) is greater, and K_ER_ (Ca^2+^ affinity of the SERCA) is lower than in the other groups. V_ER_ in the controls of octanol- and the apically located controls of CBX-treated cells are variable and nearly evenly distributed throughout the values. K_ER_ parameters in these groups mostly take higher values (from 0.11 to 0.15 µM).

**Figure 10 ijms-24-11095-f010:**
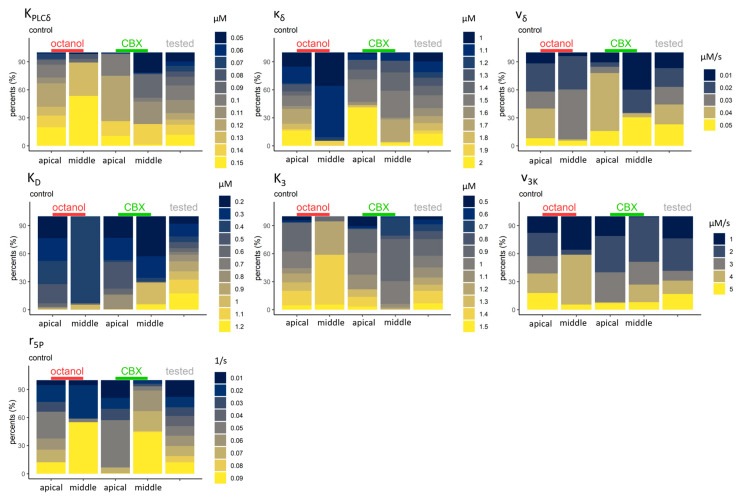
Intracellular dynamics of IP_3_. The accepted parameters of the PLC_δ_ enzyme (which plays a role in the production of IP_3_) activity in the upper row are: K_PLCδ_ is the Ca^2+^ affinity to the PLC_δ_, κ_δ_ is the inhibition constant of this enzyme activity, and v_δ_ is the maximal rate of the IP_3_ production. Higher K_PLCδ_ is present in most groups, and lower parameters are only found in the middle region in controls of CBX-treated cells. More than 50% of the accepted values for controls of octanol-treated cells from the middle region belong to the highest, 0.15 µM value, while the inhibition was also lower in this group (κ_δ_ = 1 or 1.1 µM in 94%). v_δ_ varies greatly between the groups and has uneven distribution in the different groups. The IP_3_ degradation by 3-kinases (middle row) and 5-phosphatases (third row) was also analysed. For the 3-kinases, the parameters were the Ca^2+^ (K_D_) and IP_3_ (K_3_) affinity and the maximal degradational rate (v_3K_). K_D_ mainly takes lower values, but also higher values are represented in the distribution. In contrast, mostly higher K_3_ (IP_3_ affinity) values are represented in the distribution. To compare these two values, the lowest K_D_ is connected to the highest K_3_. However, v_3K_ exhibited a nearly even distribution between the higher and lower values. The 5-phosphatase parameter r_5P_, the maximal degradation rate of IP_3_, also varied highly. Here, some tonotopic differences could be seen in the distribution of accepted parameters: the values for the apical region remain low, while they are higher for the middle region.

**Table 1 ijms-24-11095-t001:** Amplitudes, durations and AUCs of ATP-induced Ca^2+^ transients in the absence and presence of gap junction blockers in the soma of DCs in the apical and middle turns of the hemicochlea preparation. The table shows mean ± SEM (apical region: n = 8, n = 13 for octanol and CBX experiments, respectively; middle region: n = 8, n = 7 for octanol and CBX experiments, respectively) of duration, amplitude, and extrapolated AUC of the fluorescence intensity responses according to the treatment (absence (−) or presence (+) of gap junction blocker). *** *p* < 0.001, * *p* < 0.05 compared to the control, tested by the linear model, after logarithmic transformation.

Gap Junction Blocker	Treatment	Apical Region			Middle Region		
		Amplitude (dF/F_0_)	Duration (s)	AUC (dF/F0 × s)	Amplitude (dF/F_0_)	Duration (s)	AUC (dF/F0 × s)
Octanol (n.s.)	(−)	0.7 ± 0.15	42.73 ± 8.38	29.45 ± 9.58	0.66 ± 0.16	44.71 ± 13.75	34.45 ± 15.57
	(+)	0.68 ± 0.19	50.47 ± 7.04	35.89 ± 14.33	0.54 ± 0.1	36.53 ± 4.41	20.31 ± 5.7
CBX	(−)	0.35 ± 0.05	32.62 ± 4.08	10.53 ± 1.73	0.74 ± 0.18	26.57 ± 4.27	22.36 ± 7.56
	(+)	0.07 ± 0.02 ***	21.69 ± 3.56 *	1.76 ± 0.7 ***	0.13 ± 0.06 ***	12.62 ± 3.13 *	2.94 ± 1.87 ***

**Table 2 ijms-24-11095-t002:** Median and standard deviation of the reliable fitted parameter combinations for the ATP-induced Ca^2+^ transients in DCs in the apical and middle regions of the cochlea. Controls of both octanol- and CBX-treated cells were used to estimate the parameter values. Parameters that were not changed in our models: the rate of Ca^2+^ leak from the ER (r_L_ = 0.11 s^−1^), the ratio between cytosol and ER volumes (c_1_ = 0.185) and the parameters for IP_3_ receptor activation (a_2_ = 0.2 µM^−1^·s^−1^; d_1_ = 0.13 µM; d_2_ = 1.049 µM; d_3_ = 0.9434 µM; d_5_ = 0.08234 µM).). *p* shows the difference between the two regions if we compare all the mean of the parameters in all fitted combinations (n.s. non-significant differences between the parameter sets of each region, n.s. non-significant, * *p* < 0.05, ** *p* < 0.01, *** *p* < 0.001). Appendix A shows the most reliable parameters in each group and compares the parameters with the original ones used in the original De Pittá model.

Parameters	Definition	Apical Region	Middle Region	*p*
c_0_ (µM)	Total free Ca^2+^ concentration	1 ± 0.41	1 ± 0.30	**
κ_δ_ (µM)	Inhibition constant of PLC_δ_ activity	1.1 ± 0.25	1.1 ± 0.16	***
K_3_ (µM)	IP_3_ affinity of IP_3_-3-kinase	1.2 ± 0.15	1.2 ± 0.15	n.s.
K_π_ (µM)	Ca^2+^ affinity of PKC	0.6 ± 0.21	0.8 ± 0.11	***
K_D_ (µM)	Ca^2+^ affinity of IP_3_-3-kinase	0.4 ± 0.13	0.4 ± 0.19	***
K_ER_ (µM)	SERCA Ca^2+^ affinity	0.11 ± 0.024	0.11 ± 0.026	n.s.
K_p_ (µM)	Ca^2+^/PKC-dependent inhibition factor	13 ± 2.7	14 ± 1.09	***
K_PLCδ_ (µM)	Ca^2+^ affinity of PLC_δ_	0.14 ± 0.014	0.13 ± 0.018	***
K_R_ (µM)	ATP affinity of the P2Y receptor	1.3 ± 0.31	1.6 ± 0.37	***
$\bar{r_{5P}}$ (1·s^−1^)	Maximal rate of degradation by inositol-5-phosphatase	0.06 ± 0.032	0.09 ± 0.035	***
r_C_ (1·s^−1^)	Maximal rate of Ca^2+^ induced Ca^2+^ release	5 ± 0.74	4 ± 1.18	***
$\bar{v_{3K}}$ (µM·s^−1^)	Maximal rate of degradation by IP_3_-3-kinase	2 ± 1.38	3 ± 1.42	***
$\bar{v_{\beta }}$ (µM·s^−1^)	Maximal rate of IP3 production by PLC_β_	0.4 ± 0.14	0.3 ± 0.14	***
$\bar{v_{\delta }}$ (µM·s^−1^)	Maximal rate of IP_3_ production by PLC_δ_	0.02 ± 0.07	0.03 ± 0.009	***
v_ER_ (µM·s^−1^)	Maximal rate of SERCA uptake	1.3 ± 0.19	1.3 ± 0.17	***
r_P2X_ (µM·s^−1^)	Rate of the Ca^2+^ influx through P2X receptors	0.055 ± 0.015	0.035 ± 0.019	***
v_PMCA_ (µM·s^−1^)	Maximal rate of PMCA uptake	2.5 ± 0.28	2.42 ± 0.23	***
K_PMCA_ (µM)	PMCA Ca^2+^ affinity	8.7 ± 2.88	9.6 ± 3.11	*
c_i_ (µM)	Initial intracellular Ca^2+^ concentration	0.08 ± 0.017	0.07 ± 0.016	n.s.
h	Initial fraction of active IP_3_ receptors	0.7 ± 0.14	0.7 ± 0.13	***
[IP_3_] (µM)	Initial intracellular IP_3_ concentration	0.3 ± 0.14	0.3 ± 0.13	**

**Table 3 ijms-24-11095-t003:** The parameter space was randomly sampled over 300,000 times. Parameters that were not changed in our models: the rate of Ca^2+^ leak from the ER (r_L_ = 0.11 s^−1^), the ratio between cytosol and ER volumes (c_1_ = 0.185) and the parameters for IP_3_ receptor activation (a_2_ = 0.2 µM^−1^·s^−1^; d_1_ = 0.13 µM; d_2_ = 1.049 µM; d_3_ = 0.9434 µM; d_5_ = 0.08234 µM).

Parameters	Minimum	Maximum	Steps
c_0_ (µM)	1	10	1
κ_δ_ (µM)	1	2	0.1
K_3_ (µM)	0.5	1.5	0.1
K_π_ (µM)	0.1	1	0.1
K_D_ (µM)	0.2	1.2	0.1
K_ER_ (µM)	0.05	0.15	0.01
K_p_ (µM)	5	15	1
K_PLCδ_ (µM)	0.05	0.15	0.01
K_R_ (µM)	0.1	2	0.1
$\bar{r_{5P}}$ (1 s^−1^)	0.01	0.09	0.01
r_C_ (1 s^−1^)	1	10	1
$\bar{v_{3K}}$ (µM·s^−1^)	1	5	1
$\bar{v_{\beta }}$ (µM·s^−1^)	0.1	0.5	0.01
$\bar{v_{\delta }}$ (µM·s^−1^)	0.01	0.05	0.01
v_ER_ (µM·s^−1^)	0.5	1.5	0.1
r_P2X_ (µM·s^−1^)	0.005	0.1	0.001
v_PMCA_ (µM·s^−1^)	5	15	1
K_PMCA_ (µM)	2	3	0.01
c_i_ (µM)	0.05	0.1	0.01
h	0.1	1	0.1
[IP_3_] (µM)	0.1	0.5	0.1

## Data Availability

The data presented in this study are available on request from the corresponding author.

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
