# Peer review of "Ca2+ Dynamics of Gap Junction Coupled and Uncoupled Deiters’ Cells in the Organ of Corti in Hearing BALB/c Mice"

_ijms, 2023, doi:10.3390/ijms241311095_

Round 1

Reviewer 1 Report

This was a generally well written article on a complex topic. As a clinician, I found the topic fascinating and the science seems sound, although I do not feel able to critique the more technical aspects of the work. 

I have only a few very minor suggestions:

1) In Table 1, it would be helpful to indicate the non-significance of the Octanol treatment (with an n.s.) below the top half of the graph. That way the significant values at the bottom will be more clearly asociated with the CBX.

  2) The legend in the plot at the top of Figure 6 are barely legible. Could they be made in a larger font?

3) similar difficulty making out the axes labels on Figures 8 and 9.

4. The end of the second sentence of the Discussion concludes "as we waited it". I suspect this was a typographical error (weighted?) or a mis-translation.  

Author Response

Dear Reviewer,

We appreciate the valuable time and effort you spent in reviewing our paper and providing relevant comments and suggestions to make our paper clearer and more understandable. As you suggested, we have added the non-significant sign to the octanol treatment in the table 1, and thereby clarified the significance of the CBX treatment data. We have also changed the legend size in all our figures and split the figure 4. We changed the colour of the barplots to make it more elegant and less colourful. We hope these figures will be more visible to readers. We also corrected the sentence in the discussion.

Reviewer 2 Report

In this paper the authors applied a calcium indicator to a single Deiters cell in either the middle or apical region of BALB mouse hemicochlea preps, then applied ATP.  The idea was to quantify the time variant amount of Ca++ in Deiters cells with, and without, gap junctional blockers, and to compare these at two cochlear locations.  The overall questions--how does Ca++ uptake differ by cochlear location, and how does this depend on gap junctional connection of Deiters cells--are interesting and worth answering.  However, the poor writing and an apparently inadvertent 'age' experiment confound and obscure the results.  I expand on these points below.

The writing is a huge problem.  The style of writing makes it very difficult to analyze the authors' arguments.  The problem is not merely poor English, but also an inverted sentence style that makes many sentences hard to parse.  Often, sentences appeared out of order.  The use of English shows the usual problems of misuse of article adjectives and subject/verb disagreement. 

The 'control' data for experiments examining the two gap junction blockers are quite different.  This is uncommented upon, so that the reader has to wait until later sections of the paper to learn that the experiments were conducted at different ages.  Thus, between Figures 2 and 3 there are two variables, not one, hopelessly confounding the key comparisons (gap junction blocker type and similarity of middle and apical cochlear regions) of the paper.  When they acknowledge this problem, the authors make a point of their finding regarding 'aging', which the paper does not deal with in any real sense, and all references to aging should be purged.

Since the data of Figures 2-4 are then input to the author's quantitative modelling, the mix of ages undermines the point of the modelling. 

The authors intent is to test 'global' mammalian principles of cochlear operation.  When this is attempted in mice, it should never be carried out in a single inbred strain.  I will wager that the findings would be different in other inbred strains.

The statistics in Figs 2-4 seem to assume that the data are not  normally distributed, given that the summary data are described by box-and-whisker plots.  Yet the accompanying Table presents the same data as Mean+/-SEM.  I don't see how these could both be valid.  As it is, the box-and-whisker plots are extremely hard to interpret, or to notice what kind of differences the reader is supposed to glean.  It does not help that graphs that are used to support significant differences are not marked as such within each graph.

The pictorial model of Figure 5 is poorly labeled, giving little indication of how Ca++ levels are impacted by internal stores or neighboring cells.  What is even the point of the neighboring cell in the diagram?

Table 2 should have a fifth column that is a brief label for each of the metrics.

I struggled to understand Figs 8-10 and related Supp. figures.  Perhaps other reviewers find these useful, but I thought the color choices and text related to these figs were simply not helpful.

Other comments:

Line 22: 'Strongly connected' is ambiguous.  It sounds as if the OHCs are connected by gap junctions.

Line 38: The mouse cochlea has 2.25 turns.  There is no 'middle turn', but rather there are basal, middle and apical 'locations'.  This applies to other parts of the paper as well.

Line 42: 'DCs (DCs)'?

Line 43: 'Besides' is colloquial.

Line 47: 'therapeutic treatments' with no 'a' in front.

Line 53: Development is not part of transduction.

Lines 49-50. This text is vacuous.

Line 65.  Most research has NOT been conducted on newborn mice.

Line 84: Delete text after P>14.

The text at the end of the legend in Fig. 2 states that...something...is the average of Ca++ traces.  What does this refer to, and for that matter why didn't the authors simply average the Ca++ traces they recorded instead of taking a series of readings at fixed levels?

A number of references to specific figure panels in the text and legends appear to refer to the wrong panels.

In Table 2, seven of the mean values for apical versus middle locations are exactly the same, yet significant differences are claimed?

The 'Conclusions' section does not seem to list any real conclusions.

The use of English is extremely poor to the point of obscuring the meaning of the authors' words and claims.

Author Response

Dear Reviewer,

Thank you for your valuable comments, suggestions, and questions that have improved our paper. We have rewritten our paper to get rid of the inverted sentence style and make the text more understandable. Please find our responses to your other questions, comments below:

  1. The 'control' data for experiments examining the two gap junction blockers are quite different.  This is uncommented upon, so that the reader has to wait until later sections of the paper to learn that the experiments were conducted at different ages. Thus, between Figures 2 and 3 there are two variables, not one, hopelessly confounding the key comparisons (gap junction blocker type and similarity of middle and apical cochlear regions) of the paper.  When they acknowledge this problem, the authors make a point of their finding regarding 'aging', which the paper does not deal with in any real sense, and all references to aging should be purged.:

We saw that our controls are different in the two experiments, however, the statistics show that there is no significant difference between them, and we could have used the controls as one group. We used them separately to be more precise and not miss any difference. To clarify this, we added a new paragraph to the results. In this study, we did not plan to investigate the age-dependence of ATP induced Ca2+ responses, but we tried to draw a conclusion about hearing mice.  We suggest that this age-dependency requires further investigation.

  1. The authors intent is to test 'global' mammalian principles of cochlear operation.  When this is attempted in mice, it should never be carried out in a single inbred strain.  I will wager that the findings would be different in other inbred strains.:

The reviewer is absolutely right when he said that there are differences between mice strains not only in the cochlear anatomy (which had been demonstrated by Keiler 2001), but also in hearing ability, sensitivity to noise, age-related hearing losses... etc. However, we have less information on differences in purinergic signalling in the different strains. In our laboratory, CD1 strain has also been used to investigate the ATP induced Ca2+ signalling in supporting cells (Horváth et al. 2016) as wells as C57Bl/6 mice. We did not find a significant difference between these strains in case of 100 µM ATP induced responses. Gap junction blockers have only been tested in BALB/c strain mice, but we have no information on differences in connexin expression between strains. In the future, it could be interesting to study this question. To clarify our results, we changed the title of the paper to emphasised that we used only one mice strain.

  1. The statistics in Figs 2-4 seem to assume that the data are not  normally distributed, given that the summary data are described by box-and-whisker plots.  Yet the accompanying Table presents the same data as Mean+/-SEM.  I don't see how these could both be valid. As it is, the box-and-whisker plots are extremely hard to interpret, or to notice what kind of differences the reader is supposed to glean.  It does not help that graphs that are used to support significant differences are not marked as such within each graph.:

Thank you for this comment. The data on the boxplots were the same as in the table, however, the boxplots show different values (median and range of values) than those in the table (mean and SEM). To clarify it, we changed boxplots to violinplots. As mentioned by the reviewer, not all data have a normal distribution, it is easier to see it now. We decided not to show the data with barplots (with average and SEM) because we did not want to duplicate the information - which are already duplicated because Figure 4 shows the same as in Figures 2 and 3 in a different grouping to facilitate their comparison. We also indicated the significance levels in each case.

  1. The pictorial model of Figure 5 is poorly labeled, giving little indication of how Ca++ levels are impacted by internal stores or neighboring cells.  What is even the point of the neighboring cell in the diagram?:

Thank you for pointing out that our figure is not clear. We improved both the figure and the legend.

  1. Table 2 should have a fifth column that is a brief label for each of the metrics.:

We added the 5th column to make our paper easier to understand.

  1. I struggled to understand Figs 8-10 and related Supp. figures.  Perhaps other reviewers find these useful, but I thought the color choices and text related to these figs were simply not helpful.:

Thank you for pointing out that our colours do not help with understanding. We changed to a more elegant blue-yellow scale (greyscale was also tested, but the shades were not traceable), and improved the text below.

  1. Line 22: 'Strongly connected' is ambiguous.  It sounds as if the OHCs are connected by gap junctions.:

We changed it to "support".

  1. Line 38: The mouse cochlea has 2.25 turns.  There is no 'middle turn', but rather there are basal, middle and apical 'locations'.  This applies to other parts of the paper as well.:

Thank you for this comment. Turns are widely used in different papers investigating the cochlea, but this is not correct as the locations seem to be discrete in this case. However, the cells are located continuously. We changed “turns” into “regions” throughout the text.

  1. . Line 42: 'DCs (DCs)'?:

Thank you, we corrected it.

  1. Line 43: 'Besides' is colloquial.:

Thank you, we changed it to "additional".

  1. Line 47: 'therapeutic treatments' with no 'a' in front.:

Thank you, we corrected it.

  1. Line 53: Development is not part of transduction.:

Thank you, we have changed the word order of the sentence.

  1. Lines 49-50. This text is vacuous.

Thank you for pointing it out. We added these sentences to emphasize the importance of the study.  

  1. Line 65.  Most research has NOT been conducted on newborn mice.:

To study cellular mechanisms in the cochlea, P1-2 mice are generally used. Older animals are usually involved in ABR and neurological investigations. We have clarified the sentence.

  1. Line 84: Delete text after P>14.:

Thank you, we corrected it.

  1. The text at the end of the legend in Fig. 2 states that...something...is the average of Ca++ traces.  What does this refer to, and for that matter why didn't the authors simply average the Ca++ traces they recorded instead of taking a series of readings at fixed levels?:

Thank you for indicating that the sentence was not understandable, we corrected it. We chose to represent the transient responses with 7 characteristic points because the exposure of the photos was not set at the same time for each of the transients. An additional reason was that we could compare the model more easily in this case:  defining the criteria (3 out of 7 should be good) and assessing the fitness (the more reliable the model, the more points are matching) more easily. If we had chosen to compare the whole data set to the model, we would have had to deal with more data points and the automation would have been more difficult with higher computational time and capacity.

  1. A number of references to specific figure panels in the text and legends appear to refer to the wrong panels.:

Thank you, we have read carefully, and corrected all of them.

  1. In Table 2, seven of the mean values for apical versus middle locations are exactly the same, yet significant differences are claimed?:

Thank you for pointing this out, it may be misleading. The table contains the median values of all the good parameters (average values are misleading, because sometimes untested numbers would appear in this case), however the statistical tests were performed on the averages. We clarified it in the legends.

  1. The 'Conclusions' section does not seem to list any real conclusions:

Thank you, we have rewritten the conclusion.

Round 2

Reviewer 2 Report

I thank the authors for greatly improving the quality of the figures and for fixing most language problems.  But I am afraid any claims of differences by octanol versus CBX blocker groups are confounded by the differences in the CBX controls for the apical region, which in turn may reflect age differences.  Two variables have been changed between the octanol and CBX groups.  This renders any claims about either of these factors highly problematic and muddies the results of the modelling.  The authors should also have been more up-front in mentioning the age issue in the Methods, not the Discussion.  But to me, the CBX animals should be re-run, better controlling for age and seeking to eliminate the odd differences apical control data.

Author Response

Responses to reviewer 2.

Dear Reviewer,

Thank you for your valuable comment and the question that have improved our paper. Please, find our responses below.

I thank the authors for greatly improving the quality of the figures and for fixing most language problems. But I am afraid any claims of differences by octanol versus CBX blocker groups are confounded by the differences in the CBX controls for the apical region, which in turn may reflect age differences. Two variables have been changed between the octanol and CBX groups. This renders any claims about either of these factors highly problematic and muddies the results of the modelling.  The authors should also have been more up-front in mentioning the age issue in the Methods, not the Discussion. But to me, the CBX animals should be re-run, better controlling for age and seeking to eliminate the odd differences apical control data.

Thank you for this comment. According to this we went deeper in the analysis of the age and the experimental animals age distribution in the two groups which helps us to improve the discussion and the paper and hopefully, made our results more reliable.

The control groups differ in the amplitudes; however, our results did not show difference in the duration and AUC of the responses. We have made a supplementary figure (supplementary figure 1) which is depicting the data points depending on to the age of the experimental animal. A new linear model was fitted to analyse this (included the control groups, age of the experimental animals and the region of the cell). On the figure we could see how the data of the control groups are mixed. The ages where both CBX and octanol control data are present (e.g. 15-17 days in the apical region) the points are overlapping, whilst ages where only one of the control groups is presented (e.g. 18-19 days old animals in the apical region) the data were similar to the other ages. As we went deeper in the analysis, we have found some outliers which resulted in larger amplitudes of the octanol controls. However, after consultation with a statistician, we did not have reason to remove those points (no methodological artefacts or anomalies were noted in the protocols or huge differences in the experimental data).

The results in CBX controls, in the apical region, all in all, have lower variability which indicate these results are more reliable than the octanol control results.

The middle region cells were less variable in our experiments, but also there are outlier in these data sets, too, which belongs to the octanol control group (more visible in the duration).

We add all the information we had collected to the text (methods and discussion).

Round 3

Reviewer 2 Report

I thank the authors for their revisions.  The supplemental figures help a great deal.